# Evaluation of Effective Mass in InGaAsN/GaAs Quantum Wells Using Transient Spectroscopy

**DOI:** 10.3390/ma15217621

**Published:** 2022-10-30

**Authors:** Lubica Stuchlikova, Beata Sciana, Arpad Kosa, Matej Matus, Peter Benko, Juraj Marek, Martin Donoval, Wojciech Dawidowski, Damian Radziewicz, Martin Weis

**Affiliations:** 1Institute of Electronics and Photonics, Slovak University of Technology in Bratislava, Ilkovicova 3, 81219 Bratislava, Slovakia; 2Faculty of Electronics, Photonics and Microsystems, Wrocław University of Technology, 11/17 Janiszewskiego St., 50-372 Wrocław, Poland

**Keywords:** quantum well, electron effective mass, transient spectroscopy

## Abstract

Transient spectroscopies are sensitive to charge carriers released from trapping centres in semiconducting devices. Even though these spectroscopies are mostly applied to reveal defects causing states that are localised in the energy gap, these methods also sense-charge from quantum wells in heterostructures. However, proper evaluation of material response to external stimuli requires knowledge of material properties such as electron effective mass in complex structures. Here we propose a method for precise evaluation of effective mass in quantum well heterostructures. The infinite well model is successfully applied to the InGaAsN/GaAs quantum well structure and used to evaluate electron effective mass in the conduction and valence bands. The effective mass *m*/*m*_0_ of charges from the conduction band was 0.093 ± 0.006, while the charges from the conduction band exhibited an effective mass of 0.122 ± 0.018.

## 1. Introduction

The effort to decarbonise energy production and the recent energy crisis caused by the natural gas shortage are driving forces to implement renewable energy sources such as photovoltaic (PV) systems. However, even though silicon-based PV systems are the most common on the market today, the implementation of advanced semiconducting materials and structures is needed to achieve higher power conversion efficiency. Several semiconducting material families have been investigated for PV applications [1,2]; unfortunately, the single-junction solar cell cannot easily cover the whole solar spectrum. The introduction of quantum wells into PV semiconducting material is one of the envisioned approaches for improving its power conversion efficiency by expanding the absorption spectrum [3,4].

High-performance PV devices require novel materials, advanced device design, and optimised fabrication technology that provides almost defectless semiconducting materials. The semiconductor structural defects can be represented by energy states localised in the energy gap and cause photogenerated charge carrier trapping and recombination. There are two fundamental approaches to studying charge carrier nature. The photogenerated carrier lifetime before trapping and recombination is investigated by lifetime spectroscopy [5], whereas deep-level transient spectroscopy studies the charge released from traps [6]. Here should be noted that the spectroscopies characterise defects only in bulk semiconductors; the study of heterostructures always reveals the material properties of each layer as well as the influence of mutual interfaces. The application of transient spectroscopies to observe charge carriers released from the quantum well was already reported [7,8,9]; however, the analysis did not go beyond the observation of the charge release phenomenon. A more detailed analysis requires an estimation of the electron’s effective mass. Despite great progress in developing novel materials and advanced heterostructure devices, many fundamental material parameters remain not fully understood or unknown. Furthermore, ternary or quaternary semiconductors or complex heterostructures are still a great challenge for experimental methods. There are two fundamental approaches for effective mass estimation in quantum well heterostructures. The first group of methods uses optical phenomena such as angle-resolved photoemission spectroscopy (ARPES) [10], optically detected cyclotron resonance (ODCR) [11], or room-temperature optoelectronic properties (RTOP) such as reflectivity, photoluminescence or photovoltaic properties [12]. The second group of experimental techniques is based on the application of the magnetic field such as de Haas–van Alphen effect (dHvA) [13], Shubnikov-de Haas (SdH) oscillations [14], or optical quantum Hall effect (QHE) [15]. There are also other experimental techniques, such as electron energy-loss spectroscopy (EELS) using scanning electron microscopy [16]; however, their application is rare. The disadvantage of these experimental techniques is that they cannot be applied directly on an electronic device such as a diode or solar cell; hence, the results may not be fully applicable due to differences in investigated materials and/or structures. Interestingly, the pure electrical method for the evaluation of effective mass at electronic devices is still missing. As a result, the lifetime or deep-level transient spectroscopies have no other option but to apply effective mass estimated on different subjects of study.

The GaAs-based heterostructures are used as a benchmark for quantum well devices since they can be fabricated at very high reliability, and the GaAs materials are still very promising in photovoltaic applications [4,17]. Even though there are several reports on effective mass InGaAsN/GaAs or GaAsN/GaAs quantum wells, the analysis is based mostly on computer simulations [18,19] or analysis of optical properties using band anticrossing model [20,21]. As a result, there is a need for effective mass experimental estimation using an electronic device structure.

Here we report the method for evaluating the effective mass in InGaAsN/GaAs quantum wells structures using a deep-level transient spectroscopy technique. The infinite potential well model was applied to analyse the activation energies of charge carriers released from quantum wells, and the effective mass is estimated across a large number of structures.

## 2. Materials and Methods

The quantum well heterostructures were grown by atmospheric pressure metalorganic vapour phase epitaxy (APMOVPE) using a horizontal reactor (model AIX200 R&D by AIXTRON, Germany) on (100)-oriented semi-insulating GaAs or Si-doped n-type GaAs substrates. Trimethylgallium, trimethylindium, tertiarybutylhydrazine and arsine (10% AsH_3_:H_2_ mixture) were employed as the precursors, while the high-purity hydrogen was used as a carrier gas. The heterostructures consisted of 450 nm thick GaAs buffer layer, followed by triple In*_y_*Ga_1−*y*_As_1−*x*_N*_x_*/GaAs quantum wells region, and capped by 40–50 nm thick GaAs, as depicted in Figure 1 [22]. A family of 15 heterostructures of various InGaAsN thicknesses and indium/nitrogen ratios was fabricated for this study. The InGAsN layer thicknesses d varied from 6 to 19.8 nm. The nitrogen content *x* ranged from 0 to 1.2%, whereas the indium content *y* ranged from 0 to 16%. The concentric ring geometry was applied for electrode topology. Here, the voltage pulse was applied at the Pt/Ti/Pt/Au electrode, while the AuGe/Ni/Au provided the ground.

High-resolution X-ray diffractometry (HRXRD) was applied to study the structural properties of the fabricated heterostructures. The diffractograms were recorded by MRD X’Pert diffractometer (Philips, The Netherlands), using a four-crystals Bartels (220) monochromator and CuKα1 radiation (1.5406 Å). The diffractograms of the (004) symmetrical reflection were evaluated using dynamical diffraction analysis. X’Pert Epitaxy v.4.1 (PANalytical B.V., The Netherlands) software tool was employed for diffractogram modelling.

The transient spectroscopy was done by recording the capacity response on external voltage stimulus using DL800 system (BIO-RAD Micromeasurement, Mountain View, CA, USA). The temperature varied from 80 to 450 K to observe shallow states related to quantum well structures as well as the deep states representing structural defects. The excitation voltage was 0.05 V, while the charge release was observed at the reverse voltage of -0.5 V. The filling pulses of at least 3 ms were long enough to fill all states, while the transients were recorded for the period $T_{w}$ up to 1 s to observe released charge carriers. The Fourier transform analysis was employed to analyse the multiexponential transients and evaluate the Arrhenius plots [23].

## 3. Results and Discussion

All fabricated devices exhibited diode-like rectifying nature with a low leakage current in the negative voltage region, as depicted in Figure 2a, which is required for the capacitance-based transient spectroscopy. Hence, the unoccupied states were filled at 0.05 V, and capacitance transients representing the charge release were investigated at the negative voltage of −0.5 V. Note that the discrete Fourier coefficients $a_{n}$, $b_{n}$ can be estimated from each transient since the time constant τ follows the relation
(1)$$\tau ={\left(n\omega \right)}^{-1}b_{n}/a_{n},$$
where $a_{n}$, $b_{n}$ stand for the cosine and sine coefficients of *n*-th order, and the angular frequency ω is defined by the measurement period $\omega =2\pi /T_{w}$ [23,24]. Figure 2b illustrates the typical spectrum of $b_{1}$ coefficient, which denotes the first sine coefficient. The Fourier coefficient spectrum reveals the high concentration of charge released that is usually assigned to deep states of structural defects and the small concentration of charge released from quantum well states or interface charging. The deep insight can provide the Arrhenius relation for emission rate $e_{n}=\tau^{-1}$ as
(2)$$e_{n}=N_{c}\langle v_{n}\rangle \sigma_{n}\exp\left(-\left(E_{c}-E_{t}\right)/kT\right),$$
where is $N_{c}$ effective density of states at the conduction band edge, $\langle v_{n}\rangle$ is the average thermal velocity of electrons, $\sigma_{n}$ is the capture cross-section, kT is the thermal energy (k is the Boltzmann constant, and T is the thermodynamic temperature) and $E_{c}-E_{t}$ is the energy difference between the conduction band edge and the trap level localised in the energy gap. It should be mentioned here that the density of states and average thermal velocity depend on electron effective mass $m_{n}^{*}$ that can be recognised as a material parameter. As a result, the energy difference $E_{c}-E_{t}$ (also denoted as the activation energy) can be estimated precisely, whereas the capture cross-section can be evaluated only if the effective mass is known. Figure 2c depicts a typical Arrhenius plot of InGaAsN/GaAs heterostructure. Since the effective mass in InGaAsN/GaAs heterostructure is not known, we assume $m_{n}^{*}/m_{0}=1$. Interestingly, the charge released from the trap states can be easily assigned to the structural defects with activation energies of 0.694 ± 0.004 and 0.411 ± 0.001 eV. Both 0.4 and 0.7 eV trap states are well known and belong to the defects denoted as EL2 and EL5, respectively. The trap state with the activation energy of 0.411 eV is assigned to the complex defect involving the arsenic antisite, usually denoted as EL2 [25]. The trap state with the activation energy of 0.694 eV belongs to the Ga vacancy or Ga-As divacancy, known as EL5 [26].

On the other hand, the charge released from quantum wells exhibits lower activation energies and extremely low capture cross-section that points out different physical phenomena. Since the quantum well in InGaAsN/GaAs heterostructure is due to energy band offsets, as shown in Figure 1a, the slight differences in In/N ratios affect the quantum well depth. Hence, the depth was estimated assuming linear superposition of indium and nitrogen dependences of conduction (valence) band edges as illustrated in Appendix A [27]. The quality of this estimation can be verified by quantum well states occupancy analysis. For energies higher than 3kT above the Fermi energy $E_{F}$ (i.e., $E\gg E_{F}+3kT$), the distribution function f can be approximated by the Boltzmann distribution
(3)f=1/exp((E−μ)/kT),
where μ is the chemical potential that represents the quantum well depth. Since the energy level separation in quantum well always satisfy this requirement, the approximation should be valid. Furthermore, the capture cross-section estimated by the Arrhenius plot is proportional to the occupancy probability; hence, it can be used for chemical potential analysis. Figure 3 illustrates the heterostructure with x=0.32%, y=13.3%, and d=16.6 nm (evaluated in Figure 2). Interestingly, the chemical potential reaches a value of 281 ± 43 meV that is close to the value of 289 meV estimated on reported energy band offsets.

The potential distribution of InGaAsN/GaAs quantum well can be approximated as being of rectangular shape; hence, the infinity potential well can be used as a simple model [28]. Here, the electrons can occupy states with specific energy eigenvalues $E_{n}$ of the n-th quantum number that follow
(4)$$E_{n}=\frac{\hbar^{2}\pi^{2}}{2m_{n}^{*}}\left(n^{2}/d^{2}\right),$$
where is ℏ is reduced Planck constant. It should be mentioned here that the InGaAsN layer thickness d representing the quantum well width was estimated by HRXRD as shown in Appendix A. The GaAs thermal expansion coefficient has thermal dependence; however, its average value in the examined temperature range reaches a value of about 5 × 10^−6^ K^−1^ [29]. As a result, the quantum well with a width of 14 nm in a temperature difference of 300 K changes by 0.15% (i.e., 0.02 nm). This difference can be neglected since it is in the range of the experimental error. Furthermore, it is interesting to note that the quantum well eigenvalue energies in Equation (4) are linearly proportional to the square of the quantum number and quantum well width ratio, $n^{2}/d^{2}$.

As a result, it can be applied to estimate the effective mass of electrons or holes, as depicted in Figure 4. Interestingly, we can identify eigenvalues energies up to the quantum number of 4.

The conduction band effective mass of InGaAsN/GaAs quantum well was $m_{n}^{*}/m_{0}=$0.093 ± 0.006, while the valence band effective mass reached a value of $m_{p}^{*}/m_{0}=$0.122 ± 0.018. These values agree with already reported effective masses in InGaAsN/GaAs quantum well heterostructures, as it is summarised in Table 1. Note that the transient capacitance spectroscopy is capable of estimating both electron and hole effective masses in a single device, whereas commonly used experimental techniques are capable of evaluating only effective mass of major charge carriers.

## 4. Conclusions

The InGaAsN/GaAs quantum well structure was investigated using transient spectroscopy. The detailed evaluation of transients revealed the presence of charges released from deep energy states of structural defects as well as shallow states of quantum wells. The capture cross-section analysis confirmed the quantum well depth estimated by the energy band offsets. The infinite potential well model was employed to analyse the activation energies of released charges, and the effective masses of electrons in conduction and valence bands were estimated. The proposed methodology illustrates a novel approach to the evaluation of effective masses in quantum well heterostructures and offers a new roadmap for advanced device analysis.

## Figures and Tables

**Figure 1 materials-15-07621-f001:**
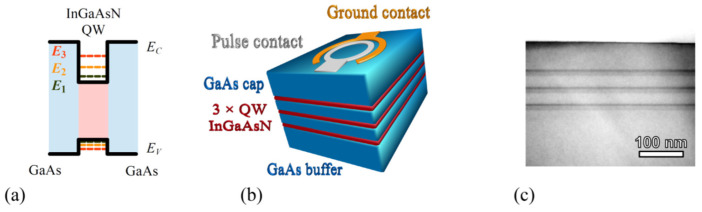
(**a**) Sketch of the energy band diagram of InGaAsN/GaAs quantum well structure. (**b**) A simplified view of the triple quantum well structure and (**c**) micrograph (transmission electron microscopy in the bright field) of fabricated heterostructure.

**Figure 2 materials-15-07621-f002:**
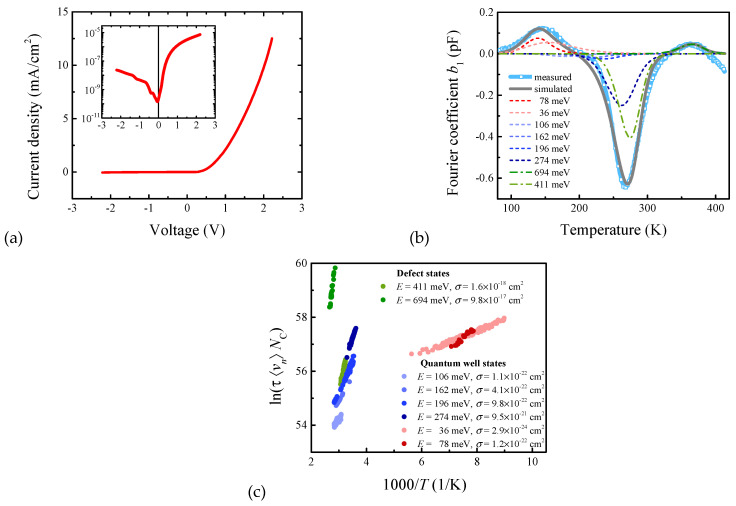
(**a**) Steady-state current-voltage characteristic of heterostructure with x=0.32%, y=13.3%, and d=16.6 nm. (**b**) The spectrum of the Fourier coefficient $b_{1}$ together with simulations of various energy states. (**c**) The Arrhenius plot.

**Figure 3 materials-15-07621-f003:**
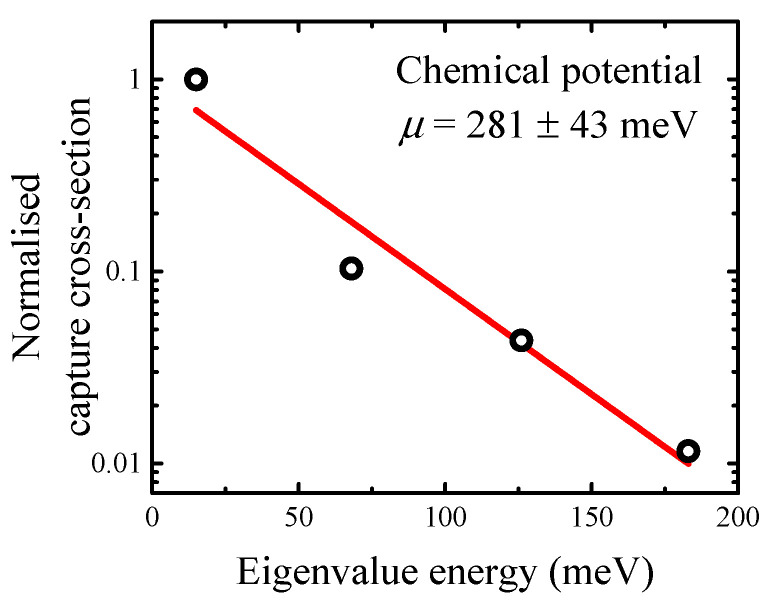
Normalised capture cross-section dependence on energy eigenvalues for heterostructure with x=0.32%, y=13.3%, and d=16.6 nm (shown in Figure 2) with the quantum well depth of 289 meV as estimated by energy band offset reports.

**Figure 4 materials-15-07621-f004:**
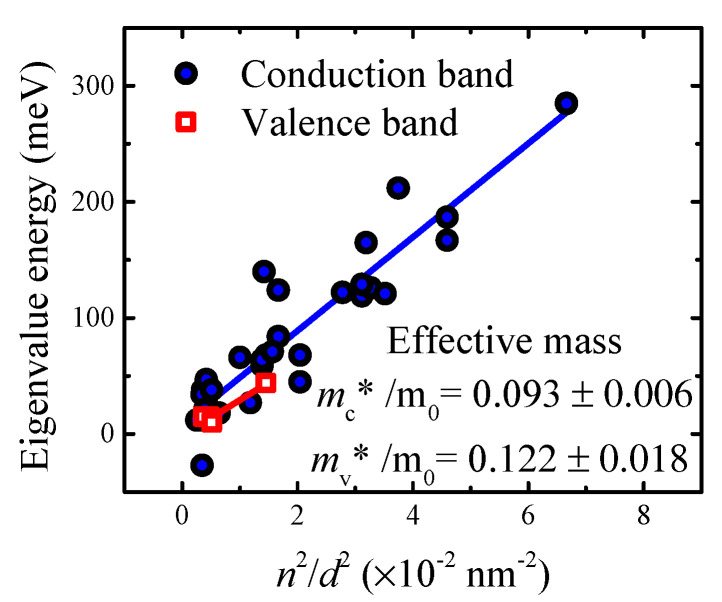
Eigenvalue energy dependence on $n^{2}/d^{2}$ ratio for conduction and valence bands. The effective mass is evaluated according to Equation (3).

**Table 1 materials-15-07621-t001:** Electron effective masses of InGaAsN/GaAs quantum well heterostructures estimated by various techniques.

$\mathrm{Effective}\;\mathrm{Mass}\;m_{n,p}^{*}/m_{0}$	Technique	Reference
0.05 ~ 0.09 (*n*)	RTOP	[12]
0.06 ~ 0.08 (*n*)	RTOP	[30]
0.077 ~ 0.078 (*n*)	QHE, SdH	[31]
0.07 ~ 0.12 (*p*)	SdH	[32]
0.07 (*n*)	ODCR	[33]
0.093 ± 0.006 (*n*)	Transient spectroscopy	This work
0.122 ± 0.018 (p)	Transient spectroscopy	This work

RTOP—room-temperature optical properties, QHE—optical quantum Hall effect, SdH—Shubnikov-de Haas oscillations, ODCR—optically detected cyclotron resonance.

## Data Availability

The data presented in this study are available in Appendix A.

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
