# Peer review of "Evaluation of Effective Mass in InGaAsN/GaAs Quantum Wells Using Transient Spectroscopy"

_materials, 2022, doi:10.3390/ma15217621_

Round 1

Reviewer 1 Report

Comments/Suggestions on the manuscript titled “Evaluation of effective mass in InGaAsN/GaAs quantum wells using transient spectroscopy

The authors have proposed a method for precise evaluation of effective mass in quantum well heterostructures using transient spectroscopy. In this report, the authors modeled an infinite quantum well structure by applying the InGaAsN/GaAs and used this to evaluate electron effective mass in the conduction and valence bands. Overall, the authors have presented a nice piece of work, which would be of interest to the research/scientific community working on solar cell developments. Nevertheless, I do have the following suggestions and comments to improve the quality of this research article:

Major comments:

1.      The introduction is too short and the objective of the review manuscript is not very clear in the introduction part. The authors should work on it.

2.      The caption of figure 4 is wrong; there is only one figure then why figure 4a and 4b, where is figure 4b?

3.      I suggest drawing a detailed comparison table between the present work and the previously reported literature, and discussing why your work is different from other authors.

4.      The conclusion should be improved in terms of the results achieved.

Minor comments:

5.      There are some grammatical errors in the manuscript, therefore the reviewer would suggest the authors revisit all the sentences and rectify the errors carefully. 

Author Response

The Authors are thankful for the careful review that helps to improve manuscript clarity. We accept all Reviewer's comments and suggestions. A detailed response follows.

Reviwer:

1. The introduction is too short and the objective of the review manuscript is not very clear in the introduction part. The authors should work on it.

Authors:

We apologize for the unclear and imperfect introduction section. We are grateful for the Reviewer's comments. The introduction was extended and experimental techniques used for effective mass estimation were discussed to illustrate state-of-the-art.

Reviewer:

2.      The caption of figure 4 is wrong; there is only one figure then why figure 4a and 4b, where is figure 4b?

Authors:

We apologize for our mistake. The Figure 3 was originally assumed to be placed as Fig. 4(b); however, to improve the discussion clarity the Figure was shifted and the Figure caption was not modified. The modified manuscript fix this issue.

Reviewer:

3.      I suggest drawing a detailed comparison table between the present work and the previously reported literature, and discussing why your work is different from other authors.

Authors:

We are grateful for this suggestion and the Table was included to illustrate the relation between present results and previously reported data.

Reviewer:

4.      The conclusion should be improved in terms of the results achieved.

Authors:

The Conclusion section was modified to point out the manuscript highlights and the bring-home message.

Reviewer:

 There are some grammatical errors in the manuscript, therefore the reviewer would suggest the authors revisit all the sentences and rectify the errors carefully. 

Authors:

We made a detailed proofreading again and language deficiencies were fixed.

Reviewer 2 Report

The authors propose a novel approach to evaluate the effective masses in quantum well heterostructures based on transient spectroscopy. The technique is applied to the InGaAsN/GaAs and used to evaluate electron and hole effective masses.

The proposal is interesting and the results are well presented.

However, I would like to suggest to the authors to better discuss in the introduction if there are to date other experimental attempts to determine the effective masses and to discuss why those are considered less successful.

Author Response

We are grateful to Reviewers comments and suggestions. All of his/her suggestions and comments were accepted. Detailed response follows.

Reviewer:

 I would like to suggest to the authors to better discuss in the introduction if there are to date other experimental attempts to determine the effective masses and to discuss why those are considered less successful.

Authors:

We apologize for the unclear and imperfect introduction section. We are grateful for the Reviewer's comments. The introduction was extended and experimental techniques used for effective mass estimation were discussed to illustrate state-of-the-art. In addition, we highlighted the outstanding nature of the proposed methodology that makes it significantly different from usually applied techniques.

Author Response

Reviewer:

  1. [Eq. (4)] The main discussion is based on the temperature dependent transient spectroscopy measurement, in which there is a 450 K difference in temperature. Should the thermal expansion of thickness “d” of InGaAsN taken into the consideration in the evaluation of effective mass.

Authors:

The Authors are grateful for the comment. The GaAs thermal expansion coefficient has thermal dependence; however, its average value in the examined temperature range reaches a value of about 5×10^-6 K^-1. As a result, the quantum well with a width (layer thickness d) of 14 nm in a temperature difference of 450 K changes by 0.25% (i.e. 0.03 nm). This difference can be neglected since it is in the range of the experimental error. Nevertheless, the appropriate discussion has been included in the modified manuscript.

--

Reviewer:

  1. [line-116] Could the origin of defect states be identified? i.e. What type of structural defects, and can it be pointed out in the HRXRD data sets? And can the width of the defect’s energy level be determined?

Authors:

The Authors are thankful for the Reviewer's interest. Since the standard defect analysis was not in the scope of the manuscript, it was omitted; however, it was included in the modified manuscript. Both 0.4 and 0.7 eV trap states are well known and belong to the defects denoted as EL2 and EL5. respectively. The trap state with the activation energy of 411 ± 1 meV is assigned to the complex defect involving the arsenic antiside (usually denoted as EL2). The trap state with the activation energy of 694 ± 4 meV belongs to the Ga vacancy or Ga-As divacancy (denoted as EL5). Since the DLTS technique has greatly higher sensitivity than the HRXRD, the concentration of defects is too small to observe them in diffractograms.

--

Reviweer:

  1. [supplementary] (a) Could authors provide a table of samples with their description of thickness, doping percentage and the according measurement number(NI--n). (b) A figures caption or description is appreciated in S3 for better understanding of four sub-figures in a data set.

Authors:

(a) The Authors are thankful for the valuable comment. The detailed table was included in modified manuscript (supplementary information). (b) the Figure captions were included.

--

Reviewer:

  1. [line-124] With the description of “At enough high temperature…”, could authors provide a range of temperature which is viable for model to fit in? [line-131] the energy offset affected by dopant percentage is estimated to be 289 meV in the case of x=0.32%, y=13.3%, and d=16.6 nm, could authors show how the number of offset is calculated?

Authors:

The Reviewer pointed out an interesting issue. The Boltzmann distribution can be considered for energies higher than 3kT above the Fermi energy. Since the energy level separation in a quantum well is at the level of 100 meV or higher, the condition is always valid. The quantum well depth was estimated using previously reported energy depths of quantum wells. The energy interpolation of experimental data was applied to have a precise evaluation.

Round 2

Reviewer 1 Report

The authors addressed all the comments carefully and enhanced their manuscript's quality; I recommend publishing this paper in the revised form.